# Conjunctive Analyses of BSA-Seq and BSR-Seq Unveil the *Msβ-GAL* and *MsJMT* as Key Candidate Genes for Cytoplasmic Male Sterility in Alfalfa (*Medicago sativa* L.)

**DOI:** 10.3390/ijms23137172

**Published:** 2022-06-28

**Authors:** Le Zhou, Yingzhe Wang, Xiaobo Xu, Dong Yan, Weijie Yu, Yifan Miao, Bo Xu

**Affiliations:** 1College of Forestry and Grassland, Jilin Agricultural University, Changchun 130118, China; zhoule228@126.com (L.Z.); xiaobo96@126.com (X.X.); yandong1925@126.com (D.Y.); ywj0102633@163.com (W.Y.); m17843096190@163.com (Y.M.); 2Institute of Agricultural Biotechnology, Jilin Academy of Agricultural Sciences (JAAS), Changchun 130119, China; yingzhe120@163.com

**Keywords:** *Medicago sativa* L., cytoplasmic male sterility, BSA-seq, BSR-seq, DEGs

## Abstract

Knowing the molecular mechanism of male sterility in alfalfa is important to utilize the heterosis more effectively. However, the molecular mechanisms of male sterility in alfalfa are still unclear. In this study, the bulked segregant analysis (BSA) and bulked segregant RNA-seq (BSR) were performed with F2 separation progeny to study the molecular mechanism of male sterility in alfalfa. The BSA-seq analysis was located in a candidate region on chromosome 5 containing 626 candidate genes which were associated with male sterility in alfalfa, while the BSR-seq analysis filtered seven candidate DEGs related to male sterility, and these candidate genes including *EF-Tu*, *β-GAL*, *CESA*, *PHGDH*, and *JMT*. The conjunctive analyses of BSR and BSA methods revealed that the genes of *Msβ-GAL* and *MsJMT* are the common detected candidate genes involved in male sterility in alfalfa. Our research provides a theory basis for further study of the molecular mechanism of male sterility in alfalfa and significant information for the genetic breeding of *Medicago sativa*.

## 1. Introduction

Plant male sterility is a genetic phenomenon in which plants fail to produce the normal functional anthers, pollen, and viable male gametes [1,2,3]. It widely exists in flowering plants and is a common biological characteristic in higher plants [4]. For the classification of male sterile type, there are genetic sterility type, physiological sterility type, and habitat sensitive sterility type caused by the expression of specific genes in a specific environment, also known as ecological sterility type [5]. Male sterility plays an important role not only in genetic breeding but also in effective utilization of heterosis. At present, cytoplasmic male sterile lines have been used in many plants, such as cotton (*Gossypium hirsutum* L.) [6], soybean (*Glycine max* L.) [7], carrot (*Daucus carota* L.) [8], corn (*Zea mays* L.) [9], onion (*Allium cepa* L.) [10], *Brassica napus* L. [11], rice (*Oryza sativa* L.) [4], sunflower (*Helianthus annuus* L.) and wheat (*Triticum aestivum* L.) [12,13].

Alfalfa (*Medicago sativa* L.) is an excellent leguminous perennial grass due to its high yield, strong disease resistance and stress tolerance, and is widely distributed globally [14,15]. As the first choice of forage resources for the development of the grass and livestock industry, the study of breeding in Alfalfa (*Medicago sativa* L.) has attracted more and more researchers since the cytoplasmic male sterile lines were discovered [16]. Although research on breeding of Alfalfa (*Medicago sativa* L.) has been going on, it is difficult to select high-quality male sterile lines and supporting maintainer lines in the process of hybrid breeding, that has obviously hindered the research on the genetic basis of fertility in alfalfa. Therefore, investigating the molecular mechanism of male sterility in alfalfa may bring new insights to solve this problem.

At present, the study of the cytological mechanism of cytoplasmic male sterility has shown that the anther morphology of fertile line and sterile line is different after microspore mitosis, and the abnormal tapetal formation of the microspore cell during meiosis is related to sporophyte deformity of the sterile line [17]. In addition, the study of the molecular mechanism of cytoplasmic male sterility indicated that *CaCWIN3* and *CaCWIN4* may be important functional genes of sucrose invertase involved in the regulation of male sterility [18]. Meanwhile, orf393 and orf288 were verified as associated with male sterility in *Allium cepa* L. and *Brassica juncea* L., respectively [19,20]. Although the molecular mechanism of male sterility has been extensively studied in other species, it is rarely reported in alfalfa. Recently, Wang and colleagues analyzed the transcriptomes of male fertile and male sterile alfalfa flower buds and identified the differentially expressed genes (DEGs) comparing two samples and found that these genes mainly participated in the pathway of circadian rhythm, transcription factors, pollen development and flavonoid biosynthesis [21]. This finding provides useful information for the study of male sterility in alfalfa, but the specific mechanism involved is still unknown.

To enhance our understanding of the molecular mechanism of male sterility in alfalfa, we selected the cytoplasmic male sterility accession of MS-GN-1A and the excellent restorer accession of MS126 as the experimental materials and obtained F2 hybridization between two parental cultivars. The BSA-seq and BSR-seq were conducted to reveal the critical mechanism of male sterility formation in alfalfa. Then, the DEGs between male fertile and male sterile alfalfa and the key candidate genes associated with male sterility were identified. Our study made a deeper analysis of the molecular mechanism of cytoplasmic male sterility in alfalfa (*Medicago sativa.* L) and provided a useful theoretical basis for alfalfa breeding.

## 2. Results

### 2.1. Sterility Survey of Parental Line and F2 Segregation Population

Figure 1 shows the pollen state of male sterile line MSGN1A and restorer line MS126 and their segregation population under the microscope. As shown in Figure 1, the pollen of sterile lines MSGN1A and sterile plants in the F2 segregation population were loosely arranged, and most of the pollen were deformed, withered, and the pollen wall was broken, exhibiting a transparent state after dyeing. However, the pollen of CMS line restorer plants MS126 and fertile plants in the F2 segregation population were arranged closely, and the pollen were plump and exhibited red or orange color after dyeing.

The identification of pollen fertility showed that there was no pollen in MSGN1A, the sterility rate of MSGN1A reached 100%, and the fertility rate of MS126 plants was 100%, suggesting that the characteristics of sterility and fertility in the parental line were stable.

Chi-square test was performed on the separation ratio composition of extremely sterile lines and restorer lines in 288 individuals of the F2 segregation population. The results showed that there were 272 fertile plants, while the number of extremely sterile plants was 16. The progressive significance was 0.626 and chi-square value was 0.237 (Appendix A).

### 2.2. Bulked-Segregant Analysis

#### 2.2.1. Sequencing Data Analysis of Sterile and Fertile Bulks

The DNAs of two libraries (R03(sterile) and R04(fertile)) were sequenced and used as BSA-seq analysis by the Illumina HiSeq2500 platform. A total of 509,677,842 paired-end reads were generated, approximately 238,377,744 reads and 271,300,098 reads for R03 and R04, respectively. After mapping of the clean reads to the reference genome, 49 examples of coverage for two F2 progeny bulks were generated (Table 1). The mapping quality and alignment efficiency were over 83.69% and 90.01%, respectively.

#### 2.2.2. Association Analysis of BSA-Seq Data

A total of 7,759,648 SNPs and 1,554,511 InDels were identified between two F2 sequencing bulks (R03 and R04) by using the GATK toolkit. Then, the association analysis was conducted using SNPs and InDels with different genotypes between two samples. The candidate region for sterility was located on chromosome five with an Δ(SNP-index) above the threshold (Figure 2a,b). In total, 626 candidate genes were identified in this candidate region (Table 2, Appendix A).

### 2.3. Bulked Segregant RNA-Seq Analysis

#### 2.3.1. Sequencing Data Analysis of Four Bulks

To further gain insight into the transcriptional profiling of sterility in Alfalfa (Medicago sativa L.), the bulked segregant RNA-seq (BSR-seq) analysis was performed with four sequencing bulks (T01–T04) by the Illumina HiSeq platform. A total of 38.79 Gb of clean data was obtained. The parent bulks (T01 and T02) generated 24,944,663 and 27,026,088 clean reads, and the two F2 segregation bulks (T03 and T04) generated 39,503,495 and 38,797,934 clean reads, respectively (Table 3). The Q30 of four cDNA sequencing bulks was over 93.76%. After obtaining high-quality data, the clean reads were assembled. A total of 308,392 transcripts with N50 values of 1406 and 131,663 unigenes with N50 values of 981 were obtained.

#### 2.3.2. Comparison of Gene Expression Pattern between Two Set Pairs of Sequencing Bulks

There are 5217 genes with differential expression pattern (DEGs) between parent bulks (T01-vs.-T02), with 3022 genes significantly up-regulated and 2195 significantly down-regulated (Figure 3a, Appendix A). The 2085 genes were identified between T03 and T04 bulks, which contained 1479 up-regulated genes and 606 down-regulated genes (Figure 3b, Appendix A).

To further explore the key genes involved in the regulation of cytoplasmic male sterility in Medicago sativa, the DEGs between two set pairs (T01-vs.-T02, T03-vs.-T04) were annotated to the GO (Gene Ontology), KEGG (Kyoto Encyclopedia of Gene and Genomes) public databases.

The GO annotation of DEGs in two set pairs showed that all genes were assigned to three major GO terms, where a big proportion of DEGs were identified in biological processes and enriched in 20 subclusters (Figure 4). For KEGG annotation, a total of 1469 DEGs were assigned to 115 KEGG pathways in parent bulk group (T01-vs.-T02), and the Ribosome, Carbon metabolism, Starch and sucrose metabolism, Phenylpropanoid metabolism and Biosynthesis of amino acids were the top five most enriched pathways (Figure 5a). For T03 and T04 bulks, the 537 DEGs were assigned to 91 KEGG pathways and the Starch and sucrose metabolism was the most significantly enriched pathway (Figure 5b). In addition, the pathway of Pentose and glucuronate interconversions, Ribosome, Phenylpropanoid metabolism and plant hormone signal transduction was also significantly enriched by large DGEs. These results indicated that the sterile development may be associated with various metabolic and biological process.

#### 2.3.3. Association Analysis of BSR-Seq Data

After filtering the SNPs with multiple genotypes and reads of no more than four, the high quality 78,727 SNPs were selected for association analysis. The Euclidean Distance (ED) algorithm and the SNP-index method were conducted to identify the markers with significant difference and obtain the candidate genes. As a result, 31 genes and seven genes were obtained by the SNP-index and ED algorithm, respectively (Appendix A). The GO classification showed that these genes were most enriched in cell, organelle and cell part for cell component category and catalytic activity, binding for molecular function category, and reproduction, metabolic process, and biological regulation for biological process category (Appendix A).

### 2.4. The DEGs Involved in Cytoplasmic Male Sterility

The main cause of plant sterility is abnormal anther development, and anther development is complex, involving many biological processes [22]. Studies have shown that the duplication and rearrangement of mitochondrial genomes can lead to cytoplasmic male sterility in plants [23,24]. Moreover, the carbohydrate metabolism and plant hormone signal transduction pathways as energy supply play important roles in pollen development. To observe the genes related to sterility participating in these three pathways in alfalfa, the differentially expressed genes (DEGs) annotated in these three pathways were analyzed. The sequencing data showed that the c101056.graph_c0 (SAUR-like auxin-responsive family protein, SAURs), c102778.graph_c0 (Indole-3-acetic acid-amido synthetase, GH3.3), c116678.graph_c0 (indole-3-acetic acid-amido synthetase, GH3.5-like), c90443.graph_c0 (BZIP transcription factor, BZIP) participating in Plant hormone signal transduction (ko04075) were up-regulated in T04, and the c108656.graph_c0 (brassino-steroid insensitive 1, BRI1), c102842.graph_c0 (Ethylene-responsive transcription factor, ERF) were down-regulated in T04 (Table 4, Figure 6a). SAURs, GH3.3 and GH3.5 are all early auxin responsive genes that can regulate a variety of biological processes including plant reproduction. GH3 plays an important role in regulating auxin balance in plants [25], and the experiment of situ hybridization of GH3 in soybean showed that GH3 transcription factor could be expressed in the endodermis and protoxylem, and up-regulated during development of flower and seed [26]. As an ABA signal and abiotic stress responsive gene, BZIP is involved in flower development and other biological processes in Arabidopsis thaliana [27]. BRI1 is the receptor of brassino-lactone, and mutation of this gene may lead to phenotypic changes such as plant dwarfing, leaf shape change, abnormal vascular development and abnormal male fertility [28,29]. Studies in tomato have shown that ERF family genes are richly expressed at multiple stages of fruit ripening [30], and this gene can both positively and negatively regulate the expression of ethylene response genes [31]. Although there is no direct evidence to prove the relationship between ERF and sterility, we think that the relation between sterility and ERF family genes deserves further study.

For starch and sucrose metabolism (ko00500), 10 encoding genes, c89515.graph_c0 (Pectinesterase), c116661.graph_c0 (polygalacturonase), c118955.graph_c0 (UDP-glucuronate 4-epimerase, GAE), c101072.graph_c0 (CAZy family GT8 glycosyltransferase, GAUT11), c100736.graph_c2 (UDP-glucose dehydrogenase, UGDH), c102786.graph_c0 (Sucrose synthase, SUS), c106305.graph_c0 (Endoglucanase), c107054.graph_c0 (beta-fructo-furanosidase, INV), c111356.graph_c0 (fructokinase, FRK), c99556.graph_c0 (Trehalose-phosphate phosphatase, otsB) were up-regulated in T04 while the β-amylase encoding gene, c114245.graph_c0 was down-regulated in T04 (Table 4, Figure 6). GAUT11 is a member of the GAUT family gene that can synthesize poly-galacturonic acid, the monomer of pectin complex. The mutation of GAUT11 in Arabidopsis thaliana results in decreased secretion of mucous and mucous galacturonic acid in seed coat mucous cells [32,33]. In addition, the experiment showed that the homologous genes of GAUT11, AtGAUT13 and AtGAUT14 can regulate pollen tube development [34]. Beta-fructo-furanosidase (INV) is an important sucrose hydrolase that can hydrolyze sucrose to glucose and fructose. The research shows that Beta-fructo-furanosidase is related to cell proliferation, vegetative tissue development and plant defense response, and expressed in developing fruits, leaves and roots [35].

For DNA replication (ko03030), replication factor A protein encoding genes (c84972.graph_c0, c88743.graph_c0, c96653.graph_c0, c111708.graph_c1) were identified. Among them, two genes were up-regulated, but two genes were down-regulated in T04 (Table 4). In addition, the replication factor-A carboxy-terminal domain protein encoding gene (c104110.graph_c0) and ATP-dependent DNA helicase PIF1 (c120364.graph_c1) were also up-regulated in T04. Functional annotations indicate that these genes belong to the RPA family. Replication factor A protein (RFA), also known as RPA, is a single-stranded DNA binding protein that plays an important role in controlling DNA metabolism, such as DNA replication, repair and recombination [36].

### 2.5. The Candidate Genes Involved in Cytoplasmic Male Sterility

According to the SNP-index analysis and the ED algorithm of BSR-seq data, 31 genes and seven genes were obtained, respectively. Among them, four genes were co-filtered from SNP-index analysis and the ED algorithm analysis (Table 5). The functional annotation showed that the gene of c102760.graph_c0 was elongation factor Tu family protein (EF-Tu), the c115887.graph_c0 was Jasmonate O-methyltransferase (JMT), and the other two genes (c123275.graph_c0 and c111485.graph_c0) were unknown. The expression analysis indicated that the four genes were differentially expressed in T03 and T04. The Jasmonate O-methyl-transferase encoding gene (Transcript_ID: c115887.graph_c0) participated in the pathway of jasmonic acid synthesis, and the gene was highly expressed in T01 (male parent) and T03 (sterile) bulks. The elongation factor Tu family protein (EF-Tu) encoding gene (Transcript_ID: c102760.graph_c0) was up regulated in T04 bulks. Current studies on the function of EF-Tu in plants mainly focus on its response to heat stress and plant immune response [37,38]. The EF-Tu were identified as candidate genes related to sterility in alfalfa by BSR method, it is worth further studying of the relationship between EF-Tu and male sterility in alfalfa. The main function of JMT is the esterification of methyl jasmonate into methyl jasmonate. As a volatile plant hormone, methyl jasmonate is involved in plant development and stress response [39].

Among the 31 genes filtered by SNP-index analysis, the β-GAL encoding gene (c102682.graph_c0), CESA and PHGDH encoding genes (c113540.graph_c0 and c111062.graph_c0) were also expressed differentially between T03 and T04. The expression level of β-GAL and PHGDH was higher in T04 than that in T03. The CESA was down-regulated in T04.

Among the candidate genes identified by BSR-seq, the c115887.graph_c0 (Jasmonate O-methyltransferase encoding gene) was also identified by the association analysis of BSA-seq. The sequence alignment showed that the c115887.graph_c0 was mapped on Chr5: 12,668,765–12,689,067 kb (MsG0580024998.01) (Table 6). In addition, among the 626 genes identified in candidate region by BSA-seq, the two genes (MsG0580025125.01 and MsG0580025201.01) located in chromosome 5 of 14,126,518–14,128,116 kb and 15,181,309–15,201,352 kb, were also annotated in Beta-galactosidase (Table 6). The crosstalk analysis of the two methods indicated that the *Msβ-GAL* and *MsJMT* were the common candidate genes related to male sterility in Alfalfa.

### 2.6. Validation of Quantitative Real Time PCR for Key Genes Associated with Male Sterility in Alfalfa

To further understand the expression patterns of candidate genes, we conducted real time PCR experiments. Nine DEGs were selected and detected in four samples. The qRT-PCR results showed that the relative mRNA expression of c101056.graph_c0 (SUAR), c102778.graph_c0 (GH3), c101072.graph_c0 (GAUT11), c102786.graph_c0 (SUS), c102760.graph_c0 (EF-Tu), c104110.graph_c0 (RFA) and c120364.graph_c1 (RFA) was markedly higher at the T04 compared with T03 bulk while the relative mRNA expression of c115887.graph_c0 (JMT) was lower in T04 than in T03 (Figure 7). The expression pattern of these DEGs was consistent with the sequencing data, which demonstrated the accuracy of the sequencing data (Figure 7).

## 3. Discussion

In recent years, transcriptome analysis has played an important role in plant research, and it also provides a powerful technology for the study of male sterility mechanism. The molecular mechanism studies of sterility in cotton, soybean, rice and rapeseed by transcriptome indicated that male sterility may be related to pollen wall synthesis, carbohydrate energy metabolism, transcription factor cell signal transduction and programmed cell death [40,41]. Based on transcriptome technology, a method of low-cost rapid localization of genes was developed by combination with bulked segregant analysis (BSA) evolved from the near-isogenic line (NIL), namely Bulked Segregant RNA-seq [42,43]. This technology brings new insight into solving the problem of construct NIL in alfalfa, and has broad application prospects in gene localization of specific traits and cloning of candidate genes. Meanwhile, bulked segregant RNA-Seq (BSR) combined with bulked segregant analysis (BSA) can investigate the genetic control of agronomic traits rapidly which only need the construction of a segregating population [44,45]. In this study, the conjunctive analyses of BSA-seq and BSR-seq was performed with the parental line and F2 segregation progeny to study the molecular mechanism of male sterility at genetic and transcriptional level. A total of 626 candidate genes located in chromosome 5 were identified by BSA-seq and 7 candidate genes expressed differentially were identified by BSR-seq. In addition, two genes, *MsJMT* and *Msβ-GAL* were commonly filtered in two methods associated with sterility in alfalfa. The annotation of DEGs showed that the Ribosome, Carbon metabolism, Starch and sucrose metabolism, Phenylpropanoid metabolism and Biosynthesis of amino acids, Pentose and glucuronate interconversions and plant hormone signal transduction were the most enriched pathways. Other authors should discuss the results and how they can be interpreted from the perspective of previous studies and of the working hypotheses. The findings and their implications should be discussed in the broadest context possible. Future research directions may also be highlighted.

Anther development is a precise and complex process involving the differentiation of stamen meristem, sporogenesis and development of microsporocyte [22]. During the development of microsporocyte transformation to mature pollen grains, meiosis involving DNA double strand breaks and DNA replication is the most important. To ensure genome integrity, cells have evolved a series of DNA repair mechanisms, such as DNA repair process containing *RFA* [46,47]. In our study, we identified RFA encoding genes as DEGs between fertility bulk of T03 and sterility bulk of T04, and we believe that they may contribute to cytoplasmic male sterility in alfalfa by participating in biological processes such as DNA replication, DNA repair, control of cell cycle and cell death. In pollen development, the expression changes of related genes (such as *SuS*, *FRK*, *BAM*, etc.) during the glycolysis pathway will directly limit or destroy the energy supply of the biological system, which may affect pollen fertility. Recent studies have shown that the glycolysis pathway can significantly directly affect the polarity of pollen tube growth in *Arabidopsis thaliana*, resulting in changes in fertility [48]. FRK is a member of hexokinase, the basic function in converting fructose to hexose phosphate. It was found that inhibition of hexokinase expression could affect abnormal pollen development and lead to abnormal pollination and seed abortion in rice [49]. SuS, as the main decomposing enzyme of sucrose, has important functions in organisms, such as converting sucrose into starch. Studies have shown that during seed development, more expression of SuS is beneficial to the accumulation of nutrients, and thus contributes to seed germination and growth [50,51]. In this experiment, the sterility identification of the alfalfa segregation population was conducted using an improved TTC method, the result showing that the pollen of T03 (sterility bulk) was a deformed, withered, and broken pollen wall, but the pollen of T04 (fertility bulk) was plump. Meanwhile, the SuS encoding gene was more expressed in the fertility line of alfalfa in this study. Based on these results, we speculated that the morphology of pollen, such as dry, deformed and broken pollen wall in sterile lines of alfalfa, may be caused by lack of nutrients such as starch and the incomplete development of seed coat. This conjecture is also consistent with previous research into the molecular mechanism of cytoplasmic male sterility in Alfalfa [52]. Recent studies have shown that the starch and sucrose pathway is associated with a variety of signal transduction pathways, such as the Ethylene, ABA, GA, cytokinin signal transduction, plant hormone signal transduction, and so on [53]. As a widely regulated substance in plants, hormones were reported to be associated with cytoplasmic male sterility in corn [54]. Therefore, the pathway of plant hormone signal transduction can be used as a breakthrough in studying the mechanism of cytoplasmic male sterility. In our study, we also found many differentially expressed genes involved in the plant hormone signal transduction pathways, for example, *SAUR*, *GH3*, *ABF*, *BRI1*, *ERF1/2*. We think the relation between these genes and alfalfa sterility is worth studying. We speculated that these pathways were also essential for male sterility in alfalfa, and the DEGs in these pathways were worth further study.

Plant hormones of Jasmonates (JAs), the derivatives of fatty acids, are endogenous growth regulators in higher plants [54]. The report has shown that jasmonates play an important role in regulating the development of flower organs, pollen dehiscence, flowering process and plant resistance [55]. In terms of plant fertility, jasmonates mainly participate in the regulation of anther dehiscence. The deficiency of JA synthase will lead to anther dehiscence failing, or anther dehiscence being too early or too late, thus affecting the timely release of pollen and hindering fertilization [56]. Jasmonic acid carboxyl methyl-transferase (*JMT*) is a key enzyme that promotes the conversion of JA to MeJA [57]. Studies found that the overexpression of JMT gene can significantly increase MeJA content in potato, ginseng and other plants [58]. At the same time, some studies have shown that the overexpression of *JMT* gene can decrease the yield of seed [59]. In addition, overexpression of *JMT* reduces panicle length, seed quality and seed setting rate in rice and *arabidopsis thaliana* [60]. However, there is little research on the *JMT* affect on plant fertility, but it revealed that jasmonic acid, the reaction substrate of *JMT*, plays an important role in plant flowering [61]. In tomato and arabidopsis, jasmonic acid is necessary for the later development of flower buds, so *JMT* may play a secondary role in fertility regulation through the jasmonic acid pathway [28]. In our study, the expression level of *JMT* was significantly higher in the sterile line than in the fertile line. In addition, its expression level is also significantly higher in the male parent than in the female parent. Meanwhile, after the *MsJMT* successfully expressed in tobacco, the content of Jasmonic acid was negatively correlated with the expression level of *MsJMT*, and the overexpression of *MsJMT* decreased jasmonic acid content. Therefore, this study speculated that the high expression of *MsJMT* would decrease the content of jasmonic acid in plants, and thus affect the male fertility of alfalfa.

β-GAL (b-galactosidase), namely β-D-galactoside galcagalcato-hydrolase, is widely found in animals, plants and microorganisms [17]. Plant-derived *β-Gal* belongs to the glycoside hydrolase 35 family, and can remove β-galactose residues at the non-reducing end of β-galactose branch chains in different tissues. The substrates of *β-Gal* are arabinogalactan, rhamnogalacturonoglycan I, and xyloglucan in cellulose, and β-galactose side chain of polysaccharides such as galactose lipids and glycoproteins [62]. In 1998, Smith found that this enzyme is the only glycosidase in higher plants that can lyse polysaccharides from cell walls [63]. At present, *β-GAL* genes isolated and cloned from various plants have been confirmed to play an important role in cell wall metabolism at various stages of plant growth [64]. Pollen wall is a complex structure in cell wall types, and the development of pollen wall can directly affect pollen germination and anther dehiscence [64]. Transcriptomic analysis of arabidopsis pollen showed that two *β-GAL* genes, At1g31740 and At5g20710, homologous to TOBACCO GP92, were highly expressed in microspore and early period of anther development, and three genes of At5g56870, At2g16730 and At4g35010 were highly expressed in mature pollen [65]. In our study, the expression level of *Msβ-GAL* was up-regulated in fertile bulk (T04) and down-regulated in sterile bulk (T03), indicating that suppression of the *β-GAL* results in abnormal glucose metabolism pathways in alfalfa, then affects the energy metabolism of cell wall tissues such as anther wall and pollen wall, and even the metabolism of lipid and other substances, ultimately affecting the development of pollen.

It is well known that celluloses and hemicelluloses are the important components of pollen intine [66]. The toughness and strength of cell wall were affected by the cellulose which consisted of a β-1,4-glucan long chain. The cellulose synthase complexes (CSCs) are located on the cell membrane and produce the cellulose on pollen intine [67]. The study found that the mutant of *CESA1* gene and *CESA3* gene in Arabidopsis can not distribute the cellulose in pollen intine successfully. Meanwhile, research reported that the missense mutation of *CESA9* in rice can affect the cell wall biosynthesis and plant growth [68]. In our study, the cellulose synthase encoding gene (*CESA*) was also identified as a candidate gene involved in male sterility in alfalfa. The relative expression result indicated that the expression level of *CESA* was higher in T03 bulk (sterile) than in T04. The mechanism of *CESA* in male sterility in alfalfa was worthy of further study.

The phosphorylated pathway of serine biosynthesis (PPSB) was reported as an essential pathway for embryo, pollen and root development. The first and rate limiting step of PPSB is the reaction catalyzed by the enzyme phosphoglycerate dehydrogenase (*PGDH*). Research found that the *PGDH1* knock-out single mutants display embryo lethal and male gametophyte development defects [69]. However, *PGDH2* and *PGDH3* knock-out mutants are shown to be fertile [70]. In our study, the *PGDH* encoding gene was identified as difference expressed genes (DEGs) and up-regulated in fertile bulk (T04). Thus, we suspected that the PGDH is important for male sterility in alfalfa and the biological significance of phosphoglycerate dehydrogenase in male sterility needs further study.

In addition, the *EF-Tu* was also identified as candidate gene related to the sterility in our study. Although *EF-Tu* is an indispensable synthesis extension factor of protein in organisms, there is no direct evidence that *EF-Tu* is related to plant fertility. We suspect that in the extremely sterile population of alfalfa, the expression of this gene may affect the development of pollen and other reproductive organs, which leads to the abnormal development of pollen and ultimately to male sterility. Specific research needs to be further explored.

## 4. Materials and Methods

### 4.1. Plant Materials

The cytoplasmic male sterile line MSGN1A (with Gongnong-3 as the donor parent, the maintainer line MSGN1B as the recurrent parent, backcrossed for four generations) and its restorer MS126 were crossed and produced F1 seedings, then the F2 segregation progeny was generated from F1 self-pollination. The MSGN1A, MSGN1B and MS126 were provided by the Jilin Academy of Agricultural Sciences. These materials were planted in the grass science research field of Jilin Agricultural University in China (118°02′ E–122°18′ E/42°11′ N–43°09′ N), 2017. The soil in this area has deep dark humus layer, salt content <0.1%, and alkalinity <5%, showing weak alkaline reaction. The climate of the region is cold and wet in winter and hot and rainy in summer, generally suitable for alfalfa growth. After sowing for one year, the buds of parent line plants and F2 individuals were collected, then the petals and stipules from the bud were removed and the anther retained. A total of 50 mg samples were taken from each plant and then stored in liquid nitrogen for later use. Each sample was repeated three times.

### 4.2. Fertility Identification of Alfalfa

Pollen viability of 288 F2 generation individuals was determined by improved TTC (2,3,5-triphenyl tetrazlium chloride) method [71]. The alfalfa flowers were gently squeezed with tweezers so that the pollen naturally fell on the clean glass slide. Then a drop of improved TTC solution was added to the pollen, and the pollen was evenly mixed with an anatomical needle. After covered the coverslips and staining at 25 °C–35 °C for 20–30 min, the pollen was observed under the microscope (XSP-8CA, Shanghai optical instrument factory, Shanghai, China). The active pollen always exhibits a red, light red, or orange color. The lifeless pollen is colorless.

After the pollen of 288 alfalfa were stained by improved TTC method, the state of pollen was observed, and the number of pollens was counted under the microscope (XSP-8CA, Shanghai optical instrument factory, China). Then, the pollen abortion rate was calculated. In this experiment, pollen fertility was divided into two categories: sterility and fertility. The evaluation criteria are shown in Appendix A.

### 4.3. Construction of Illumina Library of BSA-Seq

For bulked segregant analysis (BSA-seq), the two DNA pools were constructed with two F2 segregation bulks (R03 and R04). The DNA bulk of R03 was constructed by equally mixing the genomic DNA (100 ng) of 30 F2 individual plants with sterile, and the R04 bulks was constructed with 30 F2 individual plants with fertile. The two sequencing libraries were prepared according to the standard protocol of Illumina, then, sequenced on Illumina HiSeq2500 platform (Illumina, San Diego, CA, USA).

### 4.4. Analysis of BSA-Seq Data

The raw data generated from high-throughput sequencing were filtered and clean data obtained using Cutadapt software (version 1.13) and Trimmomatic (version 0.36) [72,73]. Then, the high quality clean reads were aligned to the reference genome of *Medicago sativa* L. (Zhomgmu No.1) (https://figshare.com/articles/dataset/Medicago_sativa_genome_and_annotation_files/12623960, accessed on 7 september 2020) using BWA (Burrows-Wheeler-Aligner) software (version 0.7.15-r1140) [74]. The SAMtools software (version 1.3.1) were used to transform the SAM file to BAM, remove and mask the effects of PCR duplication [75]. Finally, the variant calling of SNPs and InDels were excavated by GATK (Genome Analysis Toolkit) [76]. All of the variants were annotated by ANNOVAR software (version 2016Feb1) [77].

The Δ(SNP-index) value of each variant was calculated by QTLseqr (R package) [78]. The regression fitting of LOESS was performed for Δ(SNP-index) on the same chromosome to obtain the associated threshold. Then the candidate regions were identified.

### 4.5. cDNA Library Preparation of BSR-Seq

Four cDNA bulks were constructed with parental lines and F2 segregation individuals (T01, T02, T03, T04). T01 and T02 were male parent bulk and female parent bulk containing thre3 individual plants, respectively). T03 and T04 were extremely sterile bulk and fertile bulk containing 50 F2 individual plants, respectively. An equal amount of DNA from each sample was achieved and mixed to construct the bulk. Total RNA was isolated from anther according to the standard protocol of the RNA kit. The purity and concentration of each RNA sample was determined by the spectrophotometer NanoDrop-2000 (Thermo Scientific, Waltham, MA, USA). Then, the transcriptome sequencing was performed on the Illumina HiSeq2500 platform (Illumina, San Diego, CA, USA).

### 4.6. Analysis of BSR-Seq Data

After obtaining clean reads by removing the lower reads and adaptors, the Trinity software was assembled and utilized [79]. Then, the reads from all four libraries were aligned back to the final assembly using Bowtie2 software [80]. The gene expression level was assessed using FPKM method. Then, the differentially expressed genes (DEGs) between two set pairs of BSR-seq bulks (T01-vs.-T02, T03-vs.-T04) were analyzed by DEGSeq software [81]. The SNPs and small indels were identified by using the GATK toolkit, sliding-window analysis was performed for (SNP-index) association analysis such as BSA-Seq analysis, and all DEGs and genes involved in candidate regions were annotated to NR, Swiss-Prot, GO, COG, KOG, and KEGG databases using BLAST.

### 4.7. qRT-PCR Validation

Total RNA was extracted from the frozen samples using the RNA extraction kit. After obtaining RNA in each sample, we selected 1 μg RNA to conduct the reverse transcription experiment, then the cDNA was obtained and diluted five times. Each sample was put in the reaction system with 1 μL of diluted cDNA, 7.5 μL of 2xSG Green PCR Master Mix and 0.25 μ for each 10 μM primer. The amplication program was carried out as follows: 95 °C for 10 min, 95 ° C for 20 s, 60 °C for 30 s, 40 cycles, and 72 °C for 30 s. The actin gene was selected as the internal reference. The 2^−^^ΔΔCT^ method was used to calculate the relative expression of each gene. There were three biological replicates and three technical replicates. The specific primers used in this study are listed in Appendix A.

## 5. Conclusions

In conclusion, we performed BSA-seq and BSR-seq analyses to reveal the molecular mechanism of alfalfa male sterility. Based on the BSA-seq data, the candidate region relating to male sterility in alfalfa was located on chromosome 5. In addition, based on the BSR-seq data, the seven crucial DEGs, *EF-Tu*, *β-GAL*, *CESA*, *PHGDH*, and *JMT* were identified for male sterility in alfalfa. Global gene expression pattern analysis showed that a large percentage of DEGs were enriched to the pathway of Ribosome, Carbon metabolism, Starch and sucrose metabolism, Phenylpropanoid metabolism and Biosynthesis of amino acids, Pentose and glucuronate interconversions, and plant hormone signal transduction. Meanwhile, the *Msβ-GAL* and *MsJMT* were common filtered by BSR-seq and BSA-seq methods and we suspect that these two genes are important to male sterility in alfalfa. Our result provides a valuable resource for breeding of alfalfa (*Medicago sativa* L.).

## Figures and Tables

**Figure 1 ijms-23-07172-f001:**
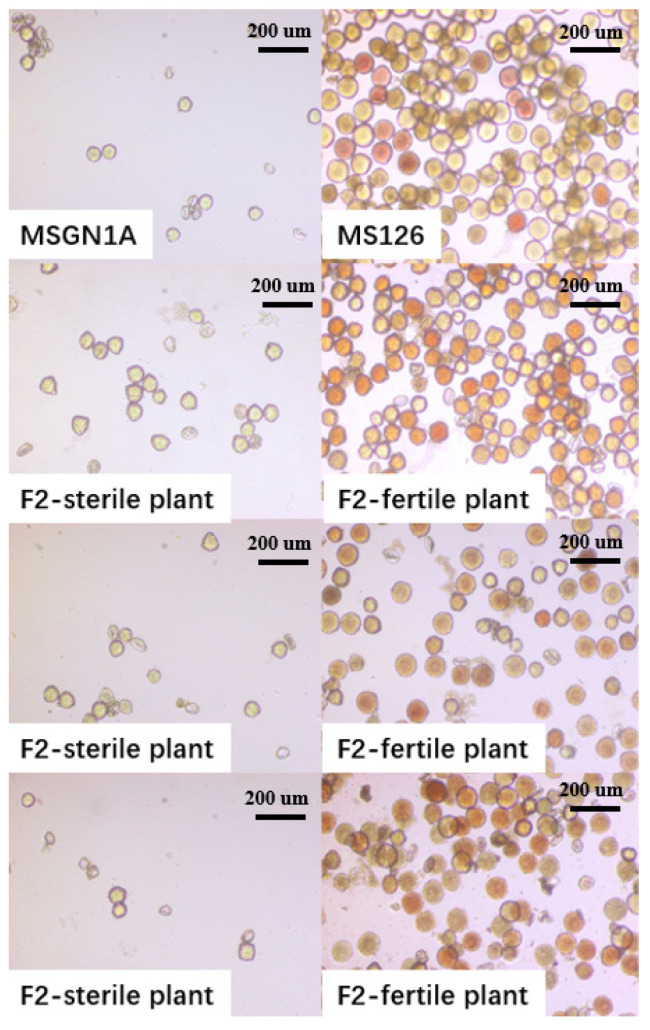
Microscope detection of pollen in alfalfa. MSGN1A is the pollen of alfalfa cytoplasmic male sterility plants, and MS126 is the pollen of recovery line plants corresponding to alfalfa cytoplasmic male sterility lines; below is the male sterility plant in the alfalfa F2 segregation population and pollen of fertile plants in the alfalfa F2 segregation population.

**Figure 2 ijms-23-07172-f002:**
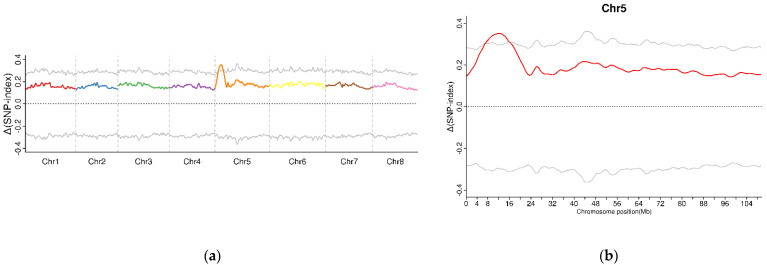
The calculation of Δ(SNP-index) values on the chromosome to identify the candidate regions associated with male sterility in alfalfa based on BSA-seq. (**a**). Candidate regions were located by the association analysis of specific SNPs between T03 and T04. (**b**). The specific candidate region on chromosome 5.

**Figure 3 ijms-23-07172-f003:**
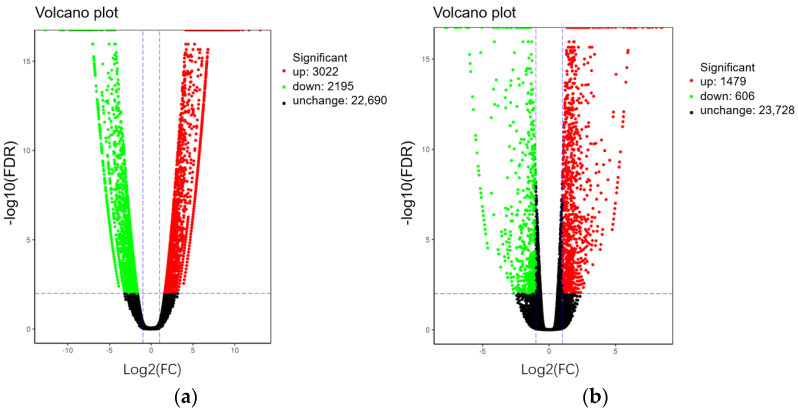
Volcano plot of the differentially expressed genes (DEG) of T01-vs.-T02 (**a**) and T04-vs.-T03 (**b**).

**Figure 4 ijms-23-07172-f004:**
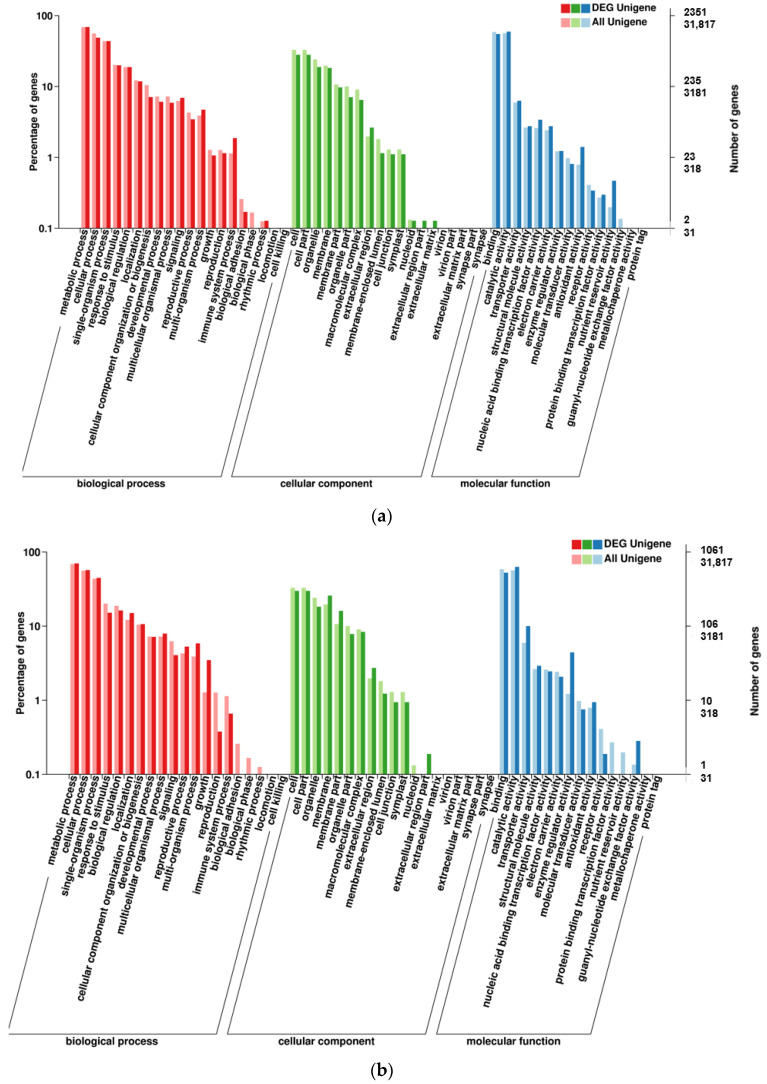
GO enrichment of DEGs in two compared groups. (**a**). T01-vs.-T02 compared group. (**b**). T04-vs.-T03 compared group.

**Figure 5 ijms-23-07172-f005:**
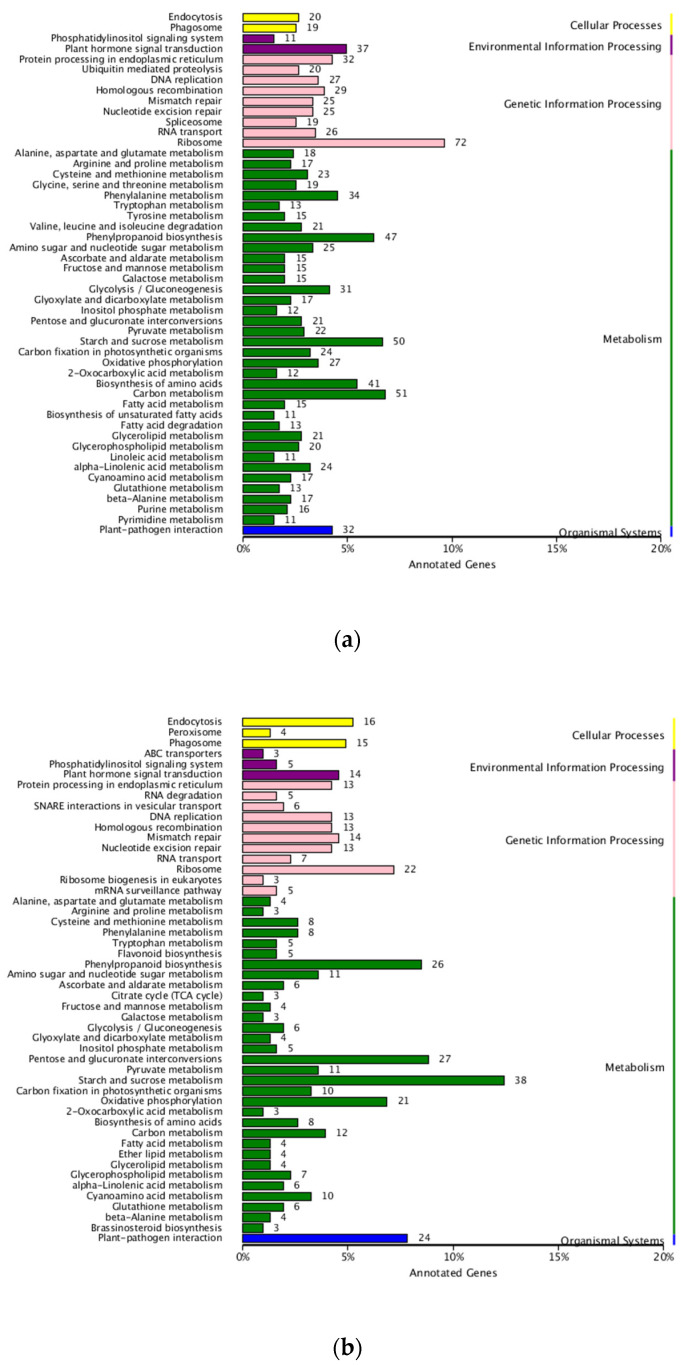
KEGG enrichment of DEGs in two compared groups. (**a**). T01-vs.-T02 compared group. (**b**). T04-vs.-T03 compared group.

**Figure 6 ijms-23-07172-f006:**
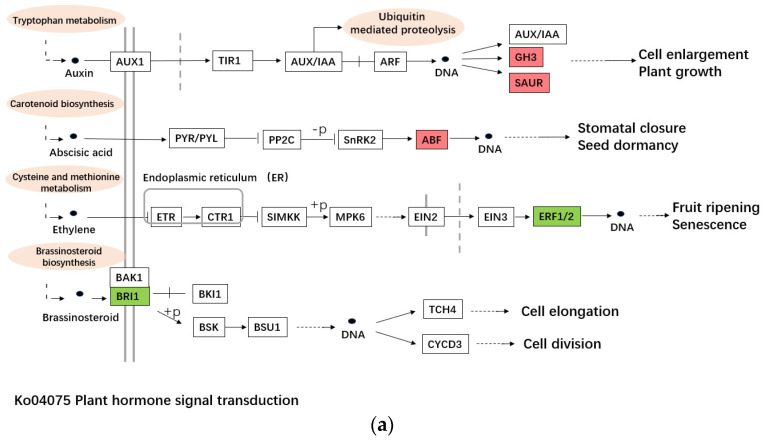
The pathways of DEGs’ participation. (**a**). The pathway of plant hormone signal transduction (ko04075). (**b**). The pathway of starch and sucrose metabolism (ko00500). (**c**). The pathway of DNA replication (ko03030).

**Figure 7 ijms-23-07172-f007:**
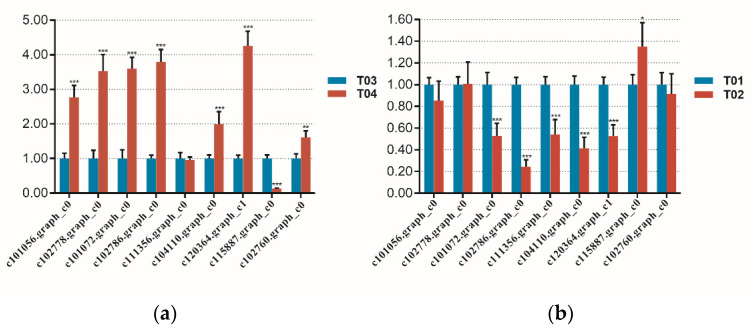
The relative mRNA expression of 9 DEGs quantified by qRT-PCR(One-way ANOVA was performed by SPSS Ststistics 17. ***, *p* < 0.01; **, *p* < 0.05; *, *p* < 0.1). (**a**) the relative mRNA expression of 9 DEGs in bulk T03 and T04. (**b**) the relative mRNA expression of 9 DEGs in bulk T01 and T02.

**Table 1 ijms-23-07172-t001:** Summary of sequencing data of BSA-seq.

Bulk	Clean Reads	Data Generated	Q30 (%)	Genome Coverage (%)	Average Depth (×)	Alignment Efficiency (%)
R03 (sterile)	238,377,744	67,766,104,095	94.29	83.69	45.92	90.01
R04 (fertile)	271,300,098	77,009,673,959	93.82	84.53	52.12	91.03

**Table 2 ijms-23-07172-t002:** Candidate genomic regions identified by association analysis of BSA-seq.

Method.	Chromosome	Start (Mb)	End (Mb)	Length (Mb)	Genes Number
SNP-index	Chromosome 5	8.54	17.02	8.47	626

**Table 3 ijms-23-07172-t003:** Summary of sequencing data of BSR-seq.

Bulk	Clean Reads	Data Generated (Gb)	Q30 (%)	Mapped Reads	Mapped Efficiency
T01 (male parent (♂))	24,944,663	7.43	94.14%	17,230,648	85.05%
T02 (female parent (♀))	27,026,088	8.06	94.01%	19,151,540	84.11%
T03 (sterile)	39,503,495	11.76	93.76%	27,499,773	85.60%
T04 (fertile)	38,797,934	11.54	94.03%	27,389,445	85.18%

**Table 4 ijms-23-07172-t004:** The sterility related DEGs participating in three key pathways.

Pathways	Transcript_ID	FPKM (T03 vs. T04)		Log_2_FC (T04/T03)	NR_Annotation	Gene Name
		T03	T04			
ko 04075 (Plant hormone signal transduction)	c101056.graph_c0	22.93	77.45	1.90	SAUR-like auxin-responsive family protein (Medicago truncatula)	SAUR
	c102778.graph_c0	0.05	1.21	3.84	Indole-3-acetic acid-amido synthetase (Medicago truncatula)	GH3
	c116678.graph_c0	16.29	39.07	1.35	indole-3-acetic acid-amido synthetase (Medicago truncatula)	GH3
	c90443.graph_c0	0.21	1.25	2.47	BZIP transcription factor (Medicago truncatula)	ABF
	c108656.graph_c0	10.01	3.89	−1.18	brassinosteroid receptor(Pisum sativum)	BRI1
	c102842.graph_c0	3.9	0.74	−2.15	ethylene response factor 5 (Medicago sativa)	ERF1/2
ko 00500 (Starch and sucrose metabolism)	c89515.graph_c0	87.9	326.14	2.03	Pectinesterase (Medicago truncatula)	Pectinesterase
	c116661.graph_c0	5.48	16.73	1.74	polygalacturonase (Medicago truncatula)	polygalacturonase
	c118955.graph_c0	50.45	100.64	1.17	UDP-glucuronate 4-epimerase (Medicago truncatula)	GAE
	c101072.graph_c0	0.2	1.03	2.25	CAZy family GT8 glycosyltransferase (Medicago truncatula)	GAUT11
	c100736.graph_c2	8.76	16.46	1.06	UDP-glucose dehydrogenase (Medicago truncatula)	UGDH
	c102786.graph_c0	0.68	2.28	1.84	Sucrose synthase (Medicago truncatula)	SUS
	c106305.graph_c0	18.08	57.22	1.80	glycosyl hydrolase family 9 protein (Medicago truncatula)	Endoglucanase
	c107054.graph_c0	54.71	119.48	1.20	beta-fructofuranosidase, insoluble protein (Medicago truncatula)	INV
	c111356.graph_c0	5.08	13.78	1.63	Fructokinase (Medicago truncatula)	FRK
	c99556.graph_c0	3.14	12.18	2.09	Trehalose-phosphate phosphatase (Medicago truncatula)	otsB
	c114245.graph_c0	29.07	10.13	−1.38	Beta-amylase (Medicago truncatula)	b-amylase
ko 03030 (DNA replication)	c104110.graph_c0	0.84	2.3	1.79	replication factor-A carboxy-terminal domain protein (Medicago truncatula)	RFA
	c120364.graph_c1	2.83	5.28	1.14	ATP-dependent DNA helicase PIF1 (Medicago truncatula)	RFA
	c84972.graph_c0	0.30	1.8	5.15	Replication factor A protein (Medicago truncatula)	RFA
	c88743.graph_c0	0.20	2.41	6.54	Replication factor A protein (Medicago truncatula)	RFA
	c96653.graph_c0	10.71	2.52	−1.99	Replication factor A protein (Medicago truncatula)	RFA
	c111708.graph_c1	1.91	0.55	−1.61	Replication factor A protein (Medicago truncatula)	RFA

**Table 5 ijms-23-07172-t005:** The 7 candidate genes identified by BSR.

Transcript_ID	Regulation (T04/T03)	Gene Name	NR_Annotation
c123275.graph_c0	down	Unknow	Unknow
c111485.graph_c0	down	Unknow	Unknow
c102760.graph_c0	up	*EF-Tu*	elongation factor Tu family protein (Medicago truncatula)
c115887.graph_c0	down	*JMT*	Jasmonate O-methyltransferase (Medicago truncatula)
c102682.graph_c0	up	*β-GAL*	Beta-galactosidase (Medicago truncatula)
c113540.graph_c0	down	*CESA*	Cellulose synthase (Medicago truncatula)
c111062.graph_c0	up	*PHGDH*	Phosphoglycerate dehydrogenase (Medicago truncatula)

**Table 6 ijms-23-07172-t006:** The 3 candidate DEGs commonly existing in BAS-seq and BSR-seq methods.

Gene_Id	Chr	Start (Kb)	End (Kb)	NR_Annotation
*MsG0580024998.01* (Transcript_id: c115887.graph_c0)	Chr5	12668765	12689067	Jasmonate O-methyltransferase (Medicago truncatula)
*MsG0580025125.01*	Chr5	14126518	14128116	Beta-galactosidase (Medicago truncatula)
*MsG0580025201.01*	Chr5	15181309	15201352	Beta-galactosidase (Medicago truncatula)

## Data Availability

The datasets of BSR-seq generated during the current study are available in SAR (sequence read archive) of NCBI [Accession number: PRJNA814740]. The datasets of BSA generated in this study are available in SAR of NCBI [Accession number: PRJNA818148].

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
