# Peer review of "Conjunctive Analyses of BSA-Seq and BSR-Seq Unveil the Msβ-GAL and MsJMT as Key Candidate Genes for Cytoplasmic Male Sterility in Alfalfa (Medicago sativa L.)"

_ijms, 2022, doi:10.3390/ijms23137172_

Round 1

Reviewer 1 Report

The manuscript “Conjunctive analyses of BSA-seq and BSR-seq unveil the Msβ-GAL and MsJMT as key candidate genes for cytoplasmic male sterility in Alfalfa (Medicago sativa L.)” present an interesting manuscript and brings new elements to existing knowledge about topic.

The manuscript is prepared professionally. It includes a well-crafted abstract and an exhaustive introduction that justifies the research undertaken. The introduction points to the deficiencies in the literature on the subject. The aim is clearly defined. Modern analytical methods were used in the research. The discussion of the results is well prepared. The conclusions are well-defined. The illustrative material is appropriate.

In my opinion, the manuscript after corrections, will be suitable for publication in a journal.

Detailed comments:

Abstract-Should be enlarge a little bit. 

Introduction - The introduction is enough in my opinion. Line 34-36

Alfalfa (Medicago sativa L.) is an excellent leguminous perennial grass due to its high yield, strong disease resistance and stress tolerance and is widely distributed in the world 35 [14] needs more references and I suggested below ones. 

Luo, Z. Analysis of the effect of reasonable close planting on respiration characteristics of alfalfa(Medicago sativa L.) artificial grassland. Turk. J. Agric. For. 2021,  45 (5), 533-540. https://doi.org/10.3906/tar-2103-110 

Rest of parts are well prepared

Reviewer 2 Report

Manuscript (MS) ijms-1783602 entitled « Conjunctive analyses of BSA-seq and BSR-seq unveil the Msβ-2 GAL and MsJMT as key candidate genes for cytoplasmic male 3 sterility in Alfalfa (Medicago sativa L.) » aimed to understand the molecular mechanisms that control male sterility in alfalfa, which is an important trait usable in plant breeding. Therefore, deciphering the genetic basis of this trait in an important plant namely alfalfa is quite relevant. From this point of view, the methodological approaches such as bulked segregant analysis (BSA) and bulked segregant RNA-seq (BSR) described in this well written MS is scientifically sound and the experimental design is appropriate. Consequently, the authors identified on the chromosome 5 about 626 candidate genes related to male sterility but conjunctive analyses of BSR and BSA revealed that Msβ-GAL and MsJMT are the common genes associated with male sterility in alfalfa. Despite these significant acheivements, the paper needs modifications before it can be considered for publication in «International Journal of Molecular Sciences».

Specific comments

Please see the attached PDF file

Round 2

Reviewer 1 Report

Dear Editor,

The authors made all necessary changes and additions. I believe that the paper is now ready for submission.

Regards